# Is the Role of Hepcidin and Erythroferrone in the Pathogenesis of Beta Thalassemia the Key to Developing Novel Treatment Strategies?

**Tsz Yuen Au** [1], **Shamiram Benjamin** [1] **and Oskar Wojciech Wiśniewski** [1,2,*]

1   Faculty of Medicine, Poznan University of Medical Sciences, 10 Fredry Street, 61-701 Poznan, Poland
2   Faculty of Health Sciences, Calisia University, 4 Nowy Świat Street, 62-800 Kalisz, Poland
*   Correspondence: wisniewski.oskar@outlook.com

**Abstract:** Thalassemia is a disease of erythrocytes that varies largely on its genetic composition and associated clinical presentation. Though some patients may remain asymptomatic, those with a complicated course may experience severe anemia early in childhood, carrying into adulthood and requiring recurrent blood transfusions as a pillar of symptom management. Due to the consequences of ineffective erythropoiesis and frequent transfusions, patients with severe beta thalassemia may be subsequently susceptible to hemochromatosis. In light of the established role of hepcidin and erythroferrone in the pathogenesis of beta thalassemia, this review aims to discuss current clinical trials and studies in the field while presenting clinical implications of the *HAMP* gene polymorphisms and novel treatments. Research suggested incorporating erythroferrone and serum hepcidin testing as a part of routine workups for beta thalassemia, as they could be a predictive tool for early iron accumulation. Furthermore, ameliorating low hepcidin and high erythroferrone appeared to be crucial in treating beta thalassemia and its complications due to iron overload. Currently, hepcidin-like compounds, such as minihepcidins, LJPC-401, PTG-300, VIT-2763, and agents that promote hepcidin production by inhibiting *TMPRSS6* expression or erythroferrone, were shown to be effective in restoring iron homeostasis in preliminary studies. Moreover, the natural bioactives astragalus polysaccharide and icariin have been recently recognized as hepcidin expression inductors.

**Keywords:** beta thalassemia; erythroferrone; *HAMP* polymorphisms; hepcidin; hemochromatosis; iron homeostasis; iron metabolism

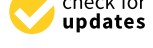



## 1. Introduction

Thalassemia is an autosomal recessive inherited hematological disorder in which functional erythrocytes and hemoglobin production are disrupted [1]. This affects both males and females and is more common within Asian, Mediterranean, and African populations, with a documented prevalence of roughly 10% compared to an approximate 1.5% prevalence of alpha and beta thalassemia carriers within the global population [1,2].

In order to better understand the pathophysiology of thalassemia and the role of hepcidin and erythroferrone within the disease process, background information on the physiological status of healthy patients should be addressed first. Normal adult hemoglobin consists of two alpha and two beta globin chains bound to iron-containing heme prosthetic groups, enabling the molecule to serve as an oxygen carrier. In patients with thalassemia, the production of either alpha or beta chains is interrupted due to deletions or mutations of the globin gene or their splice site and promoter regions. This, in turn, impairs the construction of healthy hemoglobin and, thereby, appropriate tissue oxygenation [1,3]. There exist two main types of thalassemia, aptly named based upon which globin subunit is deemed defective: alpha thalassemia and beta thalassemia.

Clinical manifestations of beta thalassemia may vary in severity based on genetic variations or zygosity, famously subtyping into "major," "intermedia," or "minor" thalassemia [3,4]. Nevertheless, ineffective erythropoiesis is a common and pivotal pathogenesis associated with thalassemic phenotypes [5]. Alpha/beta globin chain imbalance and increased oxidative stress are the main factors contributing to elevated apoptosis of erythroid progenitors in beta thalassemia [6]. In addition, impaired maturation of erythrocyte precursors and a decreased lifespan of circulating erythrocytes may exacerbate ineffective erythropoiesis [7]. As a result, chronic anemia serves as one of the hallmarks of beta thalassemia.

Beta thalassemia minor may be asymptomatic or present with mild anemia and is often diagnosed incidentally during wellness complete blood count (CBC) tests [8]. Findings on CBCs may include microcytic hypochromic anemia, which could indicate thalassemia in general, or anisopoikilocytosis, a variety of sizes and shapes, that is especially observed in beta thalassemia major [1,4]. However, in stark contrast to beta thalassemia minor, patients with beta thalassemia major typically present with severe anemia as an infant, which could be fatal if left untreated. Frequently, these patients report poor feeding in early childhood, between 6 and 24 months of age, during the transition from protective fetal hemoglobin to adult hemoglobin [4,9]. In addition to the aforementioned classic presentation, patients commonly experience irritability, failure to thrive, pallor, diarrhea, recurrent bouts of fever, and abdominal enlargement associated with hepatosplenomegaly; red blood cell transfusion is required in these patients [4]. Finally, those with beta thalassemia intermedia could encounter a myriad of medical conditions, including anemia and growth or developmental delay around 2 years of age, but blood transfusion is normally unnecessary in these cases [4].

Trying to compensate for ineffective erythropoiesis, the human body boosts the proliferation of erythroid precursors in the bone marrow, as well as promotes extramedullary hematopoiesis, mainly in the spleen and the liver [5]. Thus, hepatosplenomegaly resulting from increased erythropoietic activity is one of the signs of more severe subtypes of beta thalassemia [6]. Moreover, raised erythrocyte sequestration or destruction in the enlarged spleen may further deteriorate the patient's condition, aggravating anemia, which is a rationale for performing splenectomy as a treatment strategy in severe cases of beta thalassemia [7]. Due to persistent hemolysis partially related to hypersplenism, some patients with beta thalassemia major or intermedia are also more prone to cholelithiasis subsequent to pigment stone formation [10].

Consequently, elevated erythropoietic activity alters iron homeostasis in order to provide a sufficient pool of iron for erythrocyte formation [11]. Since the underlying cause of ineffective erythropoiesis in beta thalassemic patients remains present, the human body faces persistent excess iron absorption, which may ultimately lead to iron overload [12]. Iron accumulation and deposition may lead to the increased production of reactive oxygen species, causing subsequent tissue damage throughout the body [13]. Since the liver, heart, and endocrine glands are the most prone to gathering excess iron, the associated symptoms may, thus, include fatigue, general weakness, stomachache, weight loss, irregular heart rhythm, and endocrine abnormalities [14,15]. Furthermore, pulmonary hypertension, venous thrombosis, and osteoporosis could be complications of excessive iron loading [16]. Regular blood transfusions may be responsible for more rapid iron accumulation than in transfusion-independent patients [12]. Nevertheless, it needs to be emphasized that iron overload affects every beta thalassemic patient, although many patients could remain asymptomatic for an extended period of time unless significant organ damage occurred [12,16]. Thus, careful and regular monitoring of the amount of accumulated iron should be performed [16].

A wide array of existing research has worked to establish the key role that hepcidin serves in modulating iron metabolism in both healthy and diseased individuals—including those with beta thalassemia and even thalassemia carriers [11,17,18]. To exemplify this, cumulative studies have shown that the activation of STAT and SMAD signaling pathways may induce hepcidin expression, ultimately regulating iron concentration and filling iron stores in patients with beta thalassemia, suggesting that hepcidin may play a vital role in managing some symptoms of beta thalassemia [17]. Additionally, it has been shown that elevated erythroferrone may be responsible for inhibiting hepcidin production [19]. Therefore, in this article, we discuss the role of hepcidin and erythroferrone in the pathogenesis of beta thalassemia, paying special attention to clinical implications and novel treatment strategies.

## 2. The Physiological Role of Hepcidin and Erythroferrone in Iron Metabolism

Hepcidin is a protein that is synthesized by the liver, and it is responsible for the regulation of iron metabolism [11,20]. Hepcidin controls plasma iron levels by directly binding ferroportin, an iron export transmembrane protein present in red blood cells, macrophages, enterocytes, and hepatocytes, in order to promote its cellular lysosomal degradation [11,20]. Thus, iron absorption from enterocytes and iron mobilization within macrophages, hepatocytes, and/or bone marrow was impaired when the degradation of ferroportin occurred [11,20]. In healthy individuals, the homeostasis of hepcidin is strongly correlated to the quantity of iron in the body, both plasma and the liver. Iron excess triggers a feedback mechanism in order to regulate iron levels by elevating hepcidin concentration, whereas an increase in erythroid activity can significantly reduce hepcidin concentration so as to maintain sufficient iron supply for erythropoiesis [20,21]. Moreover, studies have shown that hepcidin rises drastically when the body is experiencing acute or chronic inflammation, which might be responsible for anemia of chronic disease(s) [11,20]. However, splenectomized patients, who are more prone to infections caused by enveloped bacteria, such as *Streptococcus pneumoniae*, *Neisseria meningitidis*, or *Haemophilus influenzae*, did not present with a significantly higher hepcidin concentration just because of the procedure [22,23]. Contrarily, the consistent anemic state in patients with beta thalassemia may work to elevate erythropoietic activity. Thus, ineffective erythropoiesis contributes to a decline in hepcidin concentration in order to ensure that plasma iron levels remain adequately elevated for the production of erythrocytes [24]. As a result, increased release of iron from intracellular stores and augmentation of intestinal absorption are observed, posing an enhanced risk of iron overload [25]. Therefore, low serum hepcidin is considered a crucial contributing factor to hemochromatosis in beta thalassemia [25,26].

Alternatively, erythroferrone is a hormone that acts as an erythroid regulator of hepcidin [27]. During enhanced erythropoietic demand or in hypoxic states, erythroferrone is produced within the bone marrow in order to downregulate the gene expression of hepcidin, encouraging increased iron availability [20]. Thus, thalassemia-related ineffective erythropoiesis acts as a significant stimulator of erythroferrone expression [28]. A myriad of studies has illustrated that erythroferrone levels are wildly elevated in individuals or animal models with beta thalassemia, leading to the suppression of hepcidin and, thereby, iron overload [19,25,29]. The increased iron loading is the major cause of morbidity and mortality in patients with beta thalassemia major [16], although its detrimental role should also not be underestimated in transfusion-independent beta thalassemia intermedia [30]. Therefore, controlling both erythroferrone and hepcidin is crucial in the modern management of beta thalassemia [29].

Interestingly, downstream effects of ineffective erythropoiesis on hepcidin synthesis were initially attributed to twisted gastrulation 1 (TWSG1) and growth differentiation factor 15 (GDF15) [31,32], albeit conflicting views arise in the literature, and future research is needed to clearly elucidate their potential [33].

### 3. The Influence of Polymorphisms of the Hepcidin Gene on the Course of Beta Thalassemia

Several studies have reported a varying severity of iron overload in beta thalassemia as a result of different single nucleotide polymorphisms in the promoter of the hepcidin-encoding *HAMP* gene [34–37]. In the studies by Andreani et al. [34] and Zarghamian et al. [35], the c.-582 A>G HAMP-P variant appeared to be associated with increased iron loading; loading of iron was calculated via measurement of serum ferritin, liver iron concentration [34], and cardiac iron deposition [35]. Since the c.-582 A>G HAMP-P variant is known to downregulate *HAMP* gene transcription [34,36], it seems that low hepcidin levels were responsible for higher iron loading activity.

Interestingly, there are discrepancies regarding the influence of the polymorphism present, dependent on the regimen of chelating therapies administered. One research group noted that a clinical significance of the c.-582 A>G HAMP-P variant could be observed in patients receiving regular chelating therapy [35], while another group evidenced an enhancement of iron loading only when chelation therapy was not administered on a regular basis [34]. Therefore, future research on a larger population to better elucidate these relationships should be conducted. If accumulating research supports these preliminary studies, there exists potential that the subpopulation of beta thalassemia patients characterized with the c.-582 A>G HAMP-P variant may take particular benefits in ameliorating low hepcidin in the future development of related treatments.

Moreover, a -72 C>T mutation in the *HAMP* promoter region was found to aggravate iron overload in a patient with beta thalassemia major [37]. Substantially elevated serum ferritin (4323 ng/mL), liver iron concentration (5230 µg/g liver dw), nontransferrin-bound iron (3.66 µM), and transferrin saturation of 110% were observed as a result of defective hepcidin molecule production [37].

Notably, polymorphisms of other genes involved in the maintenance of iron homeostasis may affect the condition of beta thalassemia patients, although it is beyond the scope of this review. Among them, *GDF15*, as well as *HFE1* and *HFE2*, typically acknowledged with hereditary hemochromatosis, were proven to escalate iron overload in beta thalassemia patients [38,39].

### 4. Recent Clinical Studies on the Influence of Hepcidin in Patients with Beta Thalassemia

Formal, controlled clinical studies were performed to study the relationship between hepcidin, erythroferrone, and beta thalassemia (Table 1). There is growing evidence showing that hepcidin suppression may be one of the vital contributors to the pathology in these patients [25,26,40]. Therefore, a thorough understanding of the pathophysiology of hepcidin and erythroferrone in beta thalassemia could benefit the future development of therapeutic approaches to diminish or relieve unpleasant symptoms caused by hemochromatosis. Ultimately, growing evidence may suggest that the upregulation of hepcidin may result in the better condition of patients, as well as a better prognosis and quality of life [40,41].

**Table 1.** Summary of clinical studies on the influence of hepcidin and erythroferrone in patients with beta thalassemia.

| Study Classification | Number of Participants | Methods | Key Findings | References |
|---|---|---|---|---|
| *Name:* Clinical study 1 <br><br> *Type:* Cross-sectional study | Test group: $n = 70$ <br> ($n = 55$: ferritin $\geq$ 1000 ng/mL; $n = 15$: ferritin < 1000 ng/mL) <br> control group: $n = 30$ | Serum concentrations of hepcidin and erythroferrone were measured along with other hematological parameters after 1 day cessation of chelation therapy | • Median serum hepcidin of the test group was statistically lower than that of the control group <br> • Median erythroferrone of the test group was statistically and significantly higher than the control group <br> • Median transferrin saturation (TS%) was almost three-fold higher in the test group | El-Rahman El-Gamal et al. [29] |
| *Name:* Clinical study 2 <br><br> *Type:* Cross-sectional study | $n = 166$ <br> ($n = 95$: beta thalassemia intermedia; $n = 49$: Hb E/β thalassemia; $n = 22$: Hb H syndromes) | Liver iron concentration (LIC), serum ferritin (SF), hepcidin, transferrin saturation (TfSat), growth differentiation factor 15 (GDF15), erythropoietin (EPO), and nontransferrin-bound iron (NTBI) were measured in thalassemia patients | • Hepcidin suppression was less pronounced in patients receiving more than 20 previous blood transfusions <br> • Hepcidin is correlated with iron turnover, indicated by transferrin saturation and nontransferrin-bound iron vs. iron storage <br> • Hepcidin/serum ferritin ratio in patients was found to be considerably lower than that of healthy individuals | Porter et al. [42] |

## 4.1. Clinical Study 1

El-Rahman El-Gamal et al., conducted a cross-sectional clinical study to compare the hematological profile of individuals with beta thalassemia and healthy people, focusing particularly on their hepcidin and erythroferrone levels within the Egyptian population [29]. The study revealed a statistically lower hepcidin level in individuals with beta thalassemia than in healthy people, whereas erythroferrone levels were at least eight-fold higher in the test group. Additionally, it was found that the median transferrin saturation was at 75% in the test group, while only 26% in the control group [29]. These results suggested that hepcidin and erythroferrone concentrations in patients with beta thalassemia are considerably different from healthy people and that the high erythroferrone and low hepcidin levels observed in the test group may be influencing transferrin saturation, serum iron concentration, and serum ferritin.

This clinical study provides a solid foundation for the understanding of the regulatory role of erythroferrone on iron profiles in beta thalassemia patients. Additionally, the findings in this research reveal erythroferrone as a potential target for the development of future treatments. Additionally, due to the observed patterns of transferrin saturation values within the clinical study and other recent papers, the incorporation of erythroferrone and serum hepcidin testing as part of routine workups for beta thalassemia may significantly increase the sensitivity of these tests and may even serve as a predictive tool for early iron accumulation in these patients [25,29,43].

Unfortunately, the study did not report a statistically significant difference in hepcidin concentration between beta thalassemic subgroups presenting with ferritin $\geq$1000 ng/mL or lower [29]. However, the *p*-value was on the very edge of reaching statistical significance ($p = 0.050$), while the studied population was small and disproportional (55 individuals with ferritin $\geq$1000 ng/mL vs. 15 participants with ferritin <1000 ng/mL). It is very likely that the recruitment of more patients to the lower ferritin subgroup could result in gaining statistical significance. Moreover, differing frequencies of blood transfusions within the tested population may influence the results as well [29].

*4.2. Clinical Study 2*

A randomized phase 2 clinical trial was conducted previously by Porter et al. [42]; a total of 166 patients participated in this study, 95 and 49 patients of whom suffered from beta thalassemia intermedia and hemoglobin E/beta thalassemia, respectively, and the remaining 22 were patients affected by hemoglobin H disease. Although considerably higher baseline hemoglobin and relatively lower erythropoietin (EPO) were observed in the patient group with hemoglobin H disease compared with the other two subgroups, all patients appeared to be anemic with an elevation in EPO. Additionally, when comparing the values among different subgroups, it could be observed that the baseline hepcidin level is significantly lower in the beta thalassemia intermedia group. In spite of the difference in the baseline level of the parameters, previous transfusion and chelation therapy were believed to be the main factors that could improve the values.

In this study, the hepcidin/serum ferritin ratio in all patients was found to be considerably lower than that of healthy individuals, regardless of disease type, serving as a strong indication of hepcidin suppression in this population [40]. Interestingly, hepcidin suppression was less pronounced in patients receiving more than 20 previous blood transfusions when compared with patients who previously received 0 to 20 transfusions [42]. This phenomenon could be explained by a reduced need for erythropoiesis and, therefore, a reduced EPO concentration, given the resolution of anemia in patients who received more transfusions; this may serve as the justification for elevated hepcidin levels [40]. Notably, more data have suggested that hepcidin is correlated with iron turnover, indicated by the presence of transferrin saturation and nontransferrin-bound iron, in lieu of preferential iron storage [42]. As a result of such findings, it is clear that single measurements of plasma hepcidin may be misleading and might not serve as the best indicator of iron overload [42]. With this in mind, it is noteworthy to perform a study on how often and when hepcidin levels should be measured in order to provide optimal healthcare to these patients.

## 5. Latest Treatment and Potential Theoretical Treatment Strategies

Ferroportin is a transmembrane protein responsible for the excretion of intracellular iron into plasma [44]. Designated "iron-exporting cells," including macrophages, duodenal enterocytes, hepatocytes, and placental syncytiotrophoblasts, act to store iron within the body. Ferroportin is present on the surface of these cells in order to readily excrete iron when required. Meanwhile, hepcidin may bind to the ferroportin of these iron-exporting cells to trigger the endocytosis and proteolysis of ferroportin, inhibiting the delivery of iron into the plasma [44]. Manolova et al., proposed the clinical application of a novel oral ferroportin inhibitor VIT-2763 for the treatment of beta thalassemia in a recent study, as it was observed that VIT-2763 not only promotes effective erythropoiesis but also rectifies the balance of myeloid precursors in the spleen in an animal model [45]. Acting competitively, VIT-2763 roughly mimics the action of hepcidin, inhibiting ferroportin by promoting its degradation, ultimately halting iron excretion into the plasma by iron-exporting cells [41,45].

A randomized phase 1 clinical trial was recently performed by Richard et al., to examine the efficacy and safety of VIT-2763 [41]. In this study, a total of 72 participants were randomly divided into two groups: single-ascending dose and multiple-ascending dose; participants in the former received one dose of up to 240 mg of either VIT-2763 or placebo, meanwhile those in the latter were administered up to 120 mg of either VIT-2763 or placebo twice daily [41]. The results support that VIT-2763 was well tolerated in healthy participants, with no major adverse effects reported. Additionally, a temporary reduction in mean serum iron concentration was observed in VIT-2763 single doses at or greater than 60 mg and all multiple doses [41]. Additionally, the hepcidin level peaked shortly after administration but appeared to be decreased over a 7-day duration in most test groups [41].

In addition to the aforementioned VIT-2763, numerous preclinical studies have suggested that therapeutic approaches that focus on the physiologic regulation of iron metabolism could be one of the future beta thalassemia treatments (Table 2). It is well supported that hepcidin plays a crucial role in regulating iron concentration in the body. Therefore, the

efficacy of minihepcidin, a hepcidin agonist composed of 7–9 *N*-terminal amino acids of hepcidin [46], should be promising by principle. Minihepcidin has been previously tested on a murine model of transfusion-dependent thalassemia with severe anemia and splenomegaly and another model of nontransfused beta thalassemia [47,48]. Studies have shown that concurrent administration of minihepcidins and red blood cell transfusions not only promotes effective erythropoiesis and improves splenomegaly but also alleviates the burden of cardiac iron overload within animal models [47,48]. These results provide a solid foundation for future clinical trials that aim to effectively treat patients with beta thalassemia [47,48].

Furthermore, LJPC-401, a synthetic human hepcidin, was tested on healthy adults to examine its effect on improving the iron profile of subjects [49]. Reduced serum iron levels and increased serum ferritin were observed in subjects, indicating that LJPC-401 may effectively regulate iron concentration. Despite the promising phase 1 results on LJPC-401 [49], a phase 2 clinical trial NCT03381833 was terminated due to a lack of efficacy as determined by interim endpoint analysis.

**Table 2.** Summary of theoretical treatments.

| Name of Therapeutic Agent | Type of Agent | Current Stage of Research |
|---|---|---|
| VIT-2763 | Oral ferroportin inhibitor | Phase 1 clinical trial performed with positive results |
| Minihepcidin | Hepcidin agonist composed of 7–9 *N*-terminal amino acids | Preclinical animal study performed with positive results |
| LJPC-401 | Synthetic human hepcidin | Phase 2 clinical trial NCT03381833 terminated due to lack of efficacy |
| PTG-300 | Hepcidin peptidomimetic | Preclinical animal study performed with positive results |
| IONIS-TMPRSS6-L | TMPRSS6 antisense oligonucleotide | Preclinical animal study performed with positive results |
| SLN124 | Small interfering RNA that reduces TMPRSS6 gene expression | Preclinical animal study performed with positive results |
| Antierythroferrone antibody | An antibody that binds to the *N*-terminal domain of erythroferrone, preventing the interaction of erythroferrone and bone morphogenetic protein | Preclinical animal study performed with positive results |
| Icariin | A natural agent that stimulates hepcidin expression | Preclinical molecular and animal study performed with positive results |
| Astragalus polysaccharide | A natural agent that stimulates hepcidin expression | Preclinical molecular and animal study performed with positive results |

TMPRSS6, transmembrane protease serine 6.

Moreover, a hepcidin peptidomimetic (PTG-300) was developed and screened in a murine model of beta thalassemia. Same as for hepcidin, the mechanism of action of PTG-300 is to reduce the saturation of serum iron and transferrin via the inhibition of ferroportin expression on cells that store or recycle iron [50]. Similar to LJPC-401, PTG-300 is designed to impede iron overload; a study on a two-mouse model of dysregulated iron homeostasis was conducted, and it was shown that the hepcidin mimetic agent PTG-300 could possibly correct dysregulated iron homeostasis and, thereby, improve disease intensity and related outcomes [50]. Comparable results were also presented at the 23rd European Hematology Association Congress by Bourne et al., who studied PTG-300 as a treatment for chronic anemia in a mouse model of beta thalassemia [51]. After 4 weeks of PTG-300 subcutaneous injections, an increase in hemoglobin, reticulocyte count, peripheral

red blood cell count, and the resolution of splenomegaly were observed, suggesting that PTG-300 might ameliorate ineffective erythropoiesis in patients with beta thalassemia [51]. Lal et al., later conducted a study to determine the pharmacodynamics of PTG-300 and its influence on transferrin saturation and serum iron [52]. Both parameters declined significantly in all testing groups differing in a regimen of PTG-300 administration, with the most sustained efficacy attributed to a twice-weekly 40 mg regimen. These preliminary results showed that PTG-300 might be capable of restoring high serum iron levels and transferrin saturation in transfusion-dependent beta thalassemia patients; however, future studies are needed to elucidate the potential of PTG-300 [52].

In addition, studies revealed that the downregulation of transmembrane protease serine 6 (TMPRSS6) was shown to upregulate hepcidin, in turn, significantly reducing serum iron and transferrin saturation in a mouse model of beta thalassemia [53,54]. The *TMPRSS6* gene instructs the production of matriptase-2, a protein that is part of a signaling cascade that regulates hepcidin [55]. Some scholars achieved reduced *TMPRSS6* activities through the utilization of specific antisense oligonucleotides [53,54], whereas others attempted to reduce *TMPRSS6* gene expression via small interfering RNA [54,56]. Despite utilizing a completely different approach, the results from the studies show promise [53,54,56,57].

Furthermore, antibodies that target the *N*-terminal domain of erythroferrone were shown to effectively improve cases of splenomegaly while reducing serum and liver iron in a mouse model of thalassemia [58]. This study suggests that the antibodies bind to the *N*-terminal domain of erythroferrone, preventing the interaction of erythroferrone and bone morphogenetic protein 6 (BMP6), therefore promoting hepcidin expression. With this in mind, these antibodies could potentially serve as a therapeutic tool for iron overload in anemic patients with beta thalassemia [58].

Lastly, some previous studies suggested using natural compounds, such as icariin and astragalus polysaccharide, to ameliorate iron disorders, as these agents boost the production of hepcidin, which absolutely warrants additional investigation [59,60]. The mechanism of action of astragalus polysaccharide consists of the elevation of interleukin-6 and p38 mitogen-activated protein kinases stimulation, both responsible for the induction of the *HAMP* gene [60]. Similarly, overexpression of hepcidin upon icariin administration is mediated by activation of the STAT3 and SMAD 1/5/8 signaling pathways [59].

Recently, the United States Food and Drug Administration (FDA) approved luspatercept as the first certified treatment for beta thalassemia [61]. Luspatercept is an erythroid maturation agent that aims to restore erythropoietic function in patients with beta thalassemia; studies showed that the medication presents with several effects, including increased hemoglobin concentration, improved morphology of erythrocytes, and prolonged lifespan of erythrocytes [61]. Moreover, the drug has the potential to reduce the need for red blood cell transfusions in patients with beta thalassemia. This new treatment shows immense promise in alleviating anemia in beta thalassemia patients; however, its effect on treating iron accumulation still remains unknown. With this in mind, it is logical to perform further studies on whether this treatment could improve iron profiles in patients with beta thalassemia. In theory, the enhancement of effective erythropoiesis may reduce the concentration of EPO, in turn increasing the serum level of hepcidin to subsequently maintain iron homeostasis within a normal physiological range.

If the aforementioned hypothesis that luspatercept may modulate iron homeostasis as a downstream consequence of its activity is proven inaccurate, it will be largely beneficial to patients if treatments that regulate iron levels were specifically developed to complement existing treatments. One viable solution is to elevate the expression of the *HAMP* gene in order to produce more hepcidin, which could be achieved by designing a *HAMP*-selective BMP, as it is known that BMP6 is responsible for promoting *HAMP* gene expression [62]. Previous studies suggested that BMP6 is key to upregulating hepcidin production, and animal studies revealed that hepcidin levels are low in BMP6 knockout mice [63]. Taking all of this into account, it is not unreasonable to assume that if a specific ligand mimicking

BMP6 were developed, hepcidin levels could be maintained at a higher concentration in these patients to prevent hemochromatosis and its associated symptoms.

## 6. Conclusions

Hepcidin is a major regulator of iron homeostasis. In the course of beta thalassemia, its concentration declines, being suppressed by increased, though ineffective, erythropoiesis. Meanwhile, elevated serum erythroferrone is observed. As a result, patients suffer from iron overload and its complications. Thus, ameliorating low hepcidin and high erythroferrone seems crucial in developing new treatment strategies. Moreover, routine measurements of serum hepcidin and erythroferrone may serve as a diagnostic tool for the early prediction of iron accumulation, albeit further studies on a clinical application are needed.

Besides the promotion of erythroid maturation by luspatercept, several compounds targeting hepcidin and erythroferrone have been proposed so far to counteract iron overload in beta thalassemia patients. They include a group of hepcidin-related agents directly mimicking the action of natural hepcidin (minihepcidins, LJPC-401, PTG-300, VIT-2763), as well as indirect hepcidin activators acting via TMPRSS6 or erythroferrone inhibition. In addition, a few natural bioactive compounds (astragalus polysaccharide, icariin) have been recently established as hepcidin expression stimulators.

Future research should elucidate the efficacy and safety of the abovementioned substances in beta thalassemia patients. Simultaneously, iron metabolism regulators should be extensively studied as potential targets of novel treatment strategies. Moreover, the identification of polymorphisms influencing iron homeostasis may serve as a background for establishing a personalized approach to the management of beta thalassemia.

**Author Contributions:** Conceptualization, O.W.W.; formal analysis, T.Y.A.; data curation, T.Y.A. and O.W.W.; writing—original draft preparation, T.Y.A., S.B. and O.W.W.; writing—review and editing, T.Y.A., S.B. and O.W.W.; supervision, O.W.W.; project administration, T.Y.A. and O.W.W. All authors have read and agreed to the published version of the manuscript.

**Funding:** This research received no external funding.

**Institutional Review Board Statement:** Not applicable.

**Informed Consent Statement:** Not applicable.

**Data Availability Statement:** Not applicable.

**Conflicts of Interest:** The authors declare no conflict of interest.

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
