# Peer review of "Is the Role of Hepcidin and Erythroferrone in the Pathogenesis of Beta Thalassemia the Key to Developing Novel Treatment Strategies?"

_thalassrep, doi:10.3390/thalassrep12030017_

Round 1

Reviewer 1 Report

The intention of the authors is clearly to explain to their readers the role of hepcidin and the clinical trials currently on the way to use hepcidin agonists to both improve erythropoiesis and iron overload. They succeed in offering the reader a clear summary of current trends. In this respect this a useful description of developments.

On the other hand it is obvious that the authors are totally familiar with the clinical aspects of thalassaemia and they need to improve on their text to avoid inaccuracies. Examples of statements that need revising include the following:

1. line 32 quote from ref 1: the documented prevalence of 10% is for both alpha and beta thalassaemia  

2. Line 51: "present with severe anaemia as adults" - no they present with severe anaemia as infants (yes after HbF is switched ) and can die if not treated before they reach the age of 5 years

3. Line 61-62: " some may suffer from haemaochromatosis" - no all will have iron overload either from blood transfusions or increased iron absorption. Also iron overload is asymptomatic until organ damage reaches a level to effect organ function.

4. In quoting reference 33 (Porter JB et al) line 197: they state that 49 patients suffer from thalassaemia minor. In fact the 49 patients of Porter et al had HbE/beta thalassaemia which can be range from intermedia to transfusion dependent. 

These are some main examples in an otherwise good paper, where hepcidin related subjects are concerned . A review is needed to iron out these inaccuracies. 

Author Response

Dear Reviewer,

the authors would like to thank you for valuable comments, which helped to improve the quality of the manuscript. All changes are marked with the blue font in the revised manuscript.

Please find our detailed revision note below.

Sincerely yours,

Authors

Reviewer 1

The intention of the authors is clearly to explain to their readers the role of hepcidin and the clinical trials currently on the way to use hepcidin agonists to both improve erythropoiesis and iron overload. They succeed in offering the reader a clear summary of current trends. In this respect this a useful description of developments.

On the other hand it is obvious that the authors are totally familiar with the clinical aspects of thalassaemia and they need to improve on their text to avoid inaccuracies. Examples of statements that need revising include the following:

  1. line 32 quote from ref 1: the documented prevalence of 10% is for both alpha and beta thalassaemia

Answer: The text has been corrected as suggested.

  1. Line 51: "present with severe anaemia as adults" - no they present with severe anaemia as infants (yes after HbF is switched) and can die if not treated before they reach the age of 5 years

Answer: The text has been corrected as suggested.

  1. Line 61-62: " some may suffer from haemaochromatosis" - no all will have iron overload either from blood transfusions or increased iron absorption. Also iron overload is asymptomatic until organ damage reaches a level to effect organ function.

Answer: The text has been corrected as suggested.

  1. In quoting reference 33 (Porter JB et al) line 197: they state that 49 patients suffer from thalassaemia minor. In fact the 49 patients of Porter et al had HbE/beta thalassaemia which can be range from intermedia to transfusion dependent.

Answer: The text has been corrected as suggested.

These are some main examples in an otherwise good paper, where hepcidin related subjects are concerned. A review is needed to iron out these inaccuracies.

Reviewer 2 Report

The review from Tsz Au et al highlights the role of hepcidin and erythroferrone in the pathogenesis of beta thalassemia and summarize the new findings of novel treatments modulating hepcidin mediated signaling.

The review is generally well written and of interest.

Part 1:

The first paragraph describes the pathophysiology of thalassemia. It’s quite basic and, most importantly, based on the title, iron overload complications and ineffective erythropoiesis are not enough described. Please, expand this part including more clinical manifestations related to these two features.

Line 45: include here also “intermedia” which is, probably, the subtype most affected by high ERFE and low Hamp.

Part 2:

Line 71-72: it is also suspected that elevated Erythroferrone… it’s not just a suspect, it was demonstrated, please rephrase and change the ref 11 referred to this sentence with a more appropriated article.

Line 78: hepcidin…. iron levels by indirectly binding ferroportin: hepcidin directly binds ferroportin, please rephrase the sentence

Line 95:… to keep serum iron at an “appropriate” level: in this case “appropriate” is not appropriate since the level of serum iron in this conditions is higher than normal, please rephrase

The role of ineffective erythropoiesis in hepcidin suppression and ERFE production is not even mention, please expand this part.

Line 105: Hamp suppression and iron overload is deleterious also in beta thal intermedia, probably more than in the TDT. Please include a sentence also on that.

Part 3:

Line 127-131: please verify the mutation and/or change the reference.

Table 1:

Clinical trial 1 and 3 cannot be consider clinical trials. Even if the study of Porter et al is based on the THALASSA trial, the paper is referring just to baseline conditions (before the treatment with deferasirox) and for this reason it’s not appropriate to refer to this study as clinical trial 3. Please change the title of the table and the Name of each study.  

Part 4.1:

Please write also the limit of the study and discuss it: they did not find a correlation between iron overload levels and hepcidin expression in beta TM patients.

Part 4.2: this is the only true clinical trial included in the table, it’s my opinion that this study should be moved to section 5, between the other trials aiming to correct iron dysregulation… In this section a table could be helpful to summarize the trials, their findings and state of the art.

Section 5:

Line 223: … could be “one of” the future of… There are other treatments which are currently ongoing trials and show promising results.

LJPC-401: clinical trial number NCT03381833: was prematurely terminated  for lack of efficacy as determined by interim endpoint analyses.

PTG-300: please include on going data from different clinical trials presented at the last EHA meeting

Author Response

Dear Reviewer,

the authors would like to thank you for valuable comments, which helped to improve the quality of the manuscript. All changes are marked with the blue font in the revised manuscript.

Please find our detailed revision note below.

Sincerely yours,

Authors

Reviewer 2

The review from Tsz Au et al highlights the role of hepcidin and erythroferrone in the pathogenesis of beta thalassemia and summarize the new findings of novel treatments modulating hepcidin mediated signaling.

The review is generally well written and of interest.

Part 1:

The first paragraph describes the pathophysiology of thalassemia. It’s quite basic and, most importantly, based on the title, iron overload complications and ineffective erythropoiesis are not enough described. Please, expand this part including more clinical manifestations related to these two features.

Answer: The first section of the manuscript has been significantly remodeled. More information on ineffective erythropoiesis and iron overload has been included and clearly linked with the pathogenesis of beta thalassemia. All changes and the new paragraphs are marked with the blue font (lines 46–89).

Line 45: include here also “intermedia” which is, probably, the subtype most affected by high ERFE and low Hamp.

Answer: The text has been corrected as suggested.

Part 2:

Line 71-72: it is also suspected that elevated Erythroferrone… it’s not just a suspect, it was demonstrated, please rephrase and change the ref 11 referred to this sentence with a more appropriated article.

Answer: The text has been corrected as suggested. In addition, the old citation has been replaced with a new reference:

[19] Kautz, L.; Jung, G.; Du, X.; Gabayan, V.; Chapman, J.; Nasoff, M.; Nemeth, E.; Ganz, T. Erythroferrone Contributes to Hepcidin Suppression and Iron Overload in a Mouse Model of β-Thalassemia. Blood 2015, 126, 2031–2037, doi:10.1182/blood-2015-07-658419.

Line 78: hepcidin…. iron levels by indirectly binding ferroportin: hepcidin directly binds ferroportin, please rephrase the sentence

Answer: The text has been corrected as suggested.

Line 95:… to keep serum iron at an “appropriate” level: in this case “appropriate” is not appropriate since the level of serum iron in this conditions is higher than normal, please rephrase

Answer: The text has been corrected as suggested.

The role of ineffective erythropoiesis in hepcidin suppression and ERFE production is not even mention, please expand this part.

Answer: In line with your suggestion, the role of ineffective erythropoiesis has been emphasized and clearly indicated in the text (lines 119-140).

Line 105: Hamp suppression and iron overload is deleterious also in beta thal intermedia, probably more than in the TDT. Please include a sentence also on that.

Answer: The text has been corrected as suggested.

Part 3:

Line 127-131: please verify the mutation and/or change the reference.

Answer: The former inappropriate citation has been replaced with the correct one:

[37] Duca L, Delbini P, Nava I, Cappellini MD, Meo A. Hepcidin mutation in a beta-thalassemia major patient with persistent severe iron overload despite chelation therapy. Intern Emerg Med. 2010 Feb;5(1):83-5. doi: 10.1007/s11739-009-0306-8. Epub 2009 Sep 12. PMID: 19756955.

Table 1:

Clinical trial 1 and 3 cannot be consider clinical trials. Even if the study of Porter et al is based on the THALASSA trial, the paper is referring just to baseline conditions (before the treatment with deferasirox) and for this reason it’s not appropriate to refer to this study as clinical trial 3. Please change the title of the table and the Name of each study. 

Answer: The phrase “clinical trial” has been replaced with “clinical study” throughout the section.

Part 4.1:

Please write also the limit of the study and discuss it: they did not find a correlation between iron overload levels and hepcidin expression in beta TM patients.

Answer: In line with your suggestions, a new paragraph regarding limitations of the clinical study 1 has been included (lines 203–210).

Part 4.2: this is the only true clinical trial included in the table, it’s my opinion that this study should be moved to section 5, between the other trials aiming to correct iron dysregulation… In this section a table could be helpful to summarize the trials, their findings and state of the art.

Answer: In line with your suggestion, we have created the Table 2 summarizing theoretical studies in section 5 and moved the paragraphs regarding the former clinical study 2 to the section 5.

Section 5:

Line 223: … could be “one of” the future of… There are other treatments which are currently ongoing trials and show promising results.

Answer: The text has been corrected as suggested.

LJPC-401: clinical trial number NCT03381833: was prematurely terminated for lack of efficacy as determined by interim endpoint analyses.

Answer: The text has been corrected as suggested.

PTG-300: please include ongoing data from different clinical trials presented at the last EHA meeting

Answer: We have included the data from the following presentation:

[53] Bourne, G.; Zhao, L.; Bhandari, A.; Frederick, B.; McMahon, J.; Tran, V.; Zhang, J.; Stephenson, A.; Taranath, R.; Tovera, J.; et al. HEPCIDIN MIMETIC PTG-300 FOR TREATMENT OF INEFFECTIVE ERYTHROPOIESIS AND CHRONIC ANEMIA IN HEMOGLOBINOPATHY DISEASES.; European Hematology Association, June 16, 2018.

We did our best but could not find any presentations regarding the use of PTG-300 in the treatment of beta thalassemia in the abstract books from the more recent EHA meetings.

Round 2

Reviewer 1 Report

This is an improved manuscript.

The only statement that I do not agree with is in line 91: iron overload will affect every patient whether symptomatic or not since tissue damage will continue; this damage must be monitored (SF, MRI) and treated before organ damage becomes irreversible. 

Line 355: measuring hepcidin and erythroferrone at clinic level still needs investigation.

Otherwise this is a good review of the role of these substances in thalassaemia and the possible role in treatment     

Author Response

Dear Reviewer,

the authors would like to thank you for valuable comments, which helped to improve the quality of the manuscript. All the new changes were made using the “track changes” mode in the revised manuscript.

Please find our detailed revision note below.

Sincerely yours,

Authors

Reviewer 1

This is an improved manuscript.

The only statement that I do not agree with is in line 91: iron overload will affect every patient whether symptomatic or not since tissue damage will continue; this damage must be monitored (SF, MRI) and treated before organ damage becomes irreversible.

Answer: Thank you for pointing out the inconsistency. The text has been corrected as suggested (lines 91-93).

Line 355: measuring hepcidin and erythroferrone at clinic level still needs investigation.

Answer: The text has been corrected as suggested (lines 366-367).

Otherwise this is a good review of the role of these substances in thalassaemia and the possible role in treatment.

Reviewer 2 Report

Authors adequately addressed all my comments. Just one last point: between the EHA2022 presentations on PTG-300 one more abstract could be included:

title: A HEPCIDIN MIMETIC, PTG-300, DEMONSTRATES PHARMACODYNAMIC EFFECTS INDICATING REDUCED IRON AVAILABILITY IN TRANSFUSION-DEPENDENT BETA-THALASSEMIA SUBJECTS

Author Response

Dear Reviewer,

the authors would like to thank you for valuable comments, which helped to improve the quality of the manuscript. All the new changes were made using the “track changes” mode in the revised manuscript.

Please find our detailed revision note below.

Sincerely yours,

Authors

Reviewer 2

Authors adequately addressed all my comments. Just one last point: between the EHA2022 presentations on PTG-300 one more abstract could be included:

title: A HEPCIDIN MIMETIC, PTG-300, DEMONSTRATES PHARMACODYNAMIC EFFECTS INDICATING REDUCED IRON AVAILABILITY IN TRANSFUSION-DEPENDENT BETA-THALASSEMIA SUBJECTS

Answer: Thank you for providing the title of EHA2022 presentation. We did our best but could not find it before. The results of the abovementioned work have been included within the paragraph referring to PTG-300 (lines 304-311). The abstract has been included in the reference list as following:

[52] Lal, A.; Voskaridou, E.; Flevari, P.; Taher, A.; Chew, L.P.; Valone, F.; Gupta, S.; Viprakasit, V. A HEPCIDIN MIMETIC, PTG-300, DEMONSTRATES PHARMACODYNAMIC EFFECTS INDICATING REDUCED IRON AVAILABILITY IN TRANSFUSION-DEPENDENT BETA-THALASSEMIA SUBJECTS
